# Practice regarding tuberculosis care among physicians at private facilities: A cross-sectional study from Vietnam

Do Minh Ngo[1]*, Ngoc Bao Doan[1], Son Nam Tran[2], Long Bao Hoang[3], Hoa Binh Nguyen[4], Vung Dang Nguyen[1]

1 Hanoi Medical University, Hanoi, Vietnam, 2 Vietnam National Hospital of Traditional Medicine, Hanoi, Vietnam, 3 Institute of Gastroenterology and Hepatology, Hanoi, Vietnam, 4 National Lung Hospital, Hanoi, Vietnam

☯ These authors contributed equally to this work.

* minhdotb@gmail.com

## Abstract

### Objectives

To evaluate the practice of TB care among physicians at private facilities.

### Methods

A cross-sectional study was conducted using questionnaires on knowledge, attitude, and practice related to TB care. The responses to these scales were used to explore latent constructs and calculate the standardized continuous scores for these domains. We described the percentages of participant's responses and explored their associated factors using multiple linear regression.

### Results

A total of 232 physicians were recruited. The most important gaps in practice included requesting chest imaging to confirm TB diagnosis (~80%), not testing HIV for confirmed active TB cases (~50%), only requesting sputum testing for MDR-TB cases (65%), only requesting follow-up examination at the end of the treatment course (64%), and not requesting sputum testing at follow-up (54%). Surgical mask was preferred to N95 respirator when examining TB patients. Prior TB training was associated with better knowledge and less stigmatizing attitude, which were associated with better practice in both TB management and precautions.

### Conclusion

There were important gaps in knowledge, attitude, and practice of TB care among private providers. Better knowledge was associated with positive attitude towards TB and better practice. Tailored training may help to address these gaps and improve the quality of TB care in the private sector.

**Data Availability Statement:** All relevant data are within the paper and its Supporting information files.

**Funding:** The author(s) received no specific funding for this work.

**Competing interests:** The authors have declared that no competing interests exist.

## Introduction

Tuberculosis (TB) is one of the world's leading ten causes of death. In 2021, the World Health Organization (WHO) estimated that there were approximately 10 million new TB cases worldwide. Among infectious diseases, tuberculosis is currently the leading cause of mortality, particularly in low- and middle-income nations. Vietnam is one of the thirty countries with the greatest incidence of tuberculosis, with 174,000 new cases and 11,000 fatalities every year. Additionally, Vietnam's burden of multidrug-resistant TB (MTD-TB) ranks eleventh among 30 countries [1].

Thirty percent of TB cases were estimated to be "missed" globally, meaning they were undetected and unrecognized by national TB programs (NTP) [2]. In many countries with a high burden of tuberculosis, the majority of missing patients are likely to seek out private facilities at least once throughout their treatment, which implies that these private facilities play a very crucial role in these countries [3]. Patients frequently seek private health care because of greater flexibility in diagnostic and treatment procedures, more convenient operating hours and locations, and less administrative burden. However, there is ample evidence that the quality of TB care in the private sector is frequently poor, and changes are urgently required. Evidence from systematic reviews of TB care quality has identified the following issues: low TB testing rates, low referral rates to the NTPs, general tendency to use antibiotics empirically and only test afterwards, knowledge and practice of private sector health workers vary widely (knowledge-to-practice gap), limited ability to support treatment compliance, and high cost of care, with half of all costs incurred before TB was diagnosed [4–7]. Additionally, a systematic review of 34 studies reported the lack of comprehensive knowledge in national TB treatment guidelines in private TB care providers. Procedures for referral, treatment monitoring, record keeping, and case holding were not routinely executed [8].

The current TB care system in Vietnam is primarily based on public health care administered by the NTP. However, many patients continue to prefer private healthcare, with 50%-70% of TB patients in Vietnam choosing private pharmacies and clinics as their initial treatment point [9–11]. Assessing the current state of TB care at private health facilities in Vietnam is challenging due to a lack of reports and records of TB cases visiting private facilities, and in many cases, the practice in these facilities is not fully legal (such as prescribing TB drugs in pharmacies, or by physicians who do not have sufficient legal expertise in TB care per the regulations regarding the scope of specialization in TB care [12]). Understanding the overall picture of knowledge, attitude, and practice in TB care at private facilities can inform policymakers and health authorities about the current status of TB care practice and the strategies to improve it. Therefore, we conducted this study to examine the practice of TB care among healthcare workers (HCWs) at private facilities and identify their association with knowledge and attitude. This will help to evaluate the potential of private sector providers in TB care, which can considerably contribute to improved outcomes of TB patients at a systemic level.

## Materials and methods

### Study design

A cross-sectional study was conducted on physicians working at private facilities in Hanoi, Ho Chi Minh City, Hai Duong, Hai Phong, Thanh Hoa, Hue from July to October 2022.

### Eligibility

Eligible participants were physicians who were available during the data collection period and working in one of the four following specialties: internal medicine, primary care, family

medicine and pulmonary medicine. TB physicians and those who were not willing to participate in the survey were excluded from the study.

## Sampling

Convenience sampling was used in this study. Owing to time and resources constraints, it was impossible to conduct probability sampling on the private facilities in the entire nation or a specific region in Vietnam. We utilized the network of private facilities currently in collaboration with the Friends for International Tuberculosis Relief (FIT) organization, as well as private facilities under the supervision of the district health centers in Hanoi, Ho Chi Minh City, Hai Duong, Hai Phong, Thanh Hoa, and Hue, and invited the physicians at those facilities to participate into the study. Study recruitment was implemented between July and October 2022, and we included all participants there were eligible and had completed the survey. Finally, a total of 232 participants at 95 private facilities were included in our analysis.

## Survey design

The survey was developed based on the implementation of existing questionnaires in literature [8, 13, 14] and the Vietnam Ministry of Health's guidelines for TB diagnosis and treatment [15] with revisions made to fit the study's setting and aims.

In addition to information on sex, age, education, and work experience, we focused on the participant's knowledge, attitude, and practice related to TB. Regarding knowledge, participants were asked to answer twelve questions about TB: (1) general knowledge, (2) clinical manifestations; (3) diagnostic and screening tests, and (4) precautions. Regarding attitude, the scale had fifteen items about participant's feelings towards TB and the TB program. Participants were asked to rate their agreement to the items using a 5-point Likert scale (0 = strongly disagree to 4 = strongly agree). Regarding practice of TB care, we asked the participants to rate their frequency of doing fourteen activities in TB management (including notification, screening/diagnosis, and patient follow up) and TB precaution. Rating also used a 5-point Likert scale (0 = never to 4 = always).

## Data collection

Because the entire study team resided in Hanoi, participants in Hanoi completed a paper survey, while participants in other cities and provinces completed an online version on Google Forms. The online survey was distributed via a shortened link to the Zalo and Viber group of physicians working at the private facilities in the network. Both versions provided information about the study and asked for consent to participate before deploying the survey. We did not collect any identifiable information, such as name and phone number. Paper survey results were entered into an electronic form designed in EpiData, and online survey results were automatically stored on Google Forms after participant hit the Submit button. The paper and online data were then cleaned and merged into a complete dataset.

## Study outcomes

This study focused on the practice of TB care among physicians at private facilities. To describe the practice at private facilities, we report the frequency of doing TB management activities, e.g., the proportion of physicians who never requested contact tracing. Items that best described the practice of participants (see the section about Exploratory Factor Analysis in Statistical analysis) were then used to calculate a standardized continuous score. We used this practice score to investigate factors that were potentially associated with practice of TB care.

## Statistical analysis

We described qualitative variables as frequencies and percentages and quantitative variables as means (standard deviation, SD) or medians (interquartile range, IQR). Horizontal stacked bar charts were used to illustrate the distribution of responses regarding attitude and practice.

Item response theory (IRT) was used to estimate the knowledge score. In short, we assumed that all knowledge True/False questions only measure one latent trait, which is the knowledge of TB care. Questions were assumed to differ from one another by their difficulty and discrimination; therefore, a 2-parameter logistic (2PL) model was used to fit the data. We first fit a model on the entire knowledge questionnaire, then eliminated questions that had poor discrimination. The final model was used to estimate the latent trait; the estimated score was standardized, with a mean 0 and variance 1.

Exploratory factor analysis (EFA) was used to estimate the attitude and practice score. Different from knowledge, we assumed that there might be some factors that can be explored from the attitude and practice scales, hence the rationale for using EFA. We used principal component analysis to preliminary assess the potential number of factors within each scale. Different criteria were used to choose the appropriate number of factors: eigenvalues >1, total variance explained >80%, eigenvalues greater than or equal to the eigenvalue at the "elbow" in the Scree plot, and observed eigenvalues greater than the eigenvalue of the same component calculated in parallel analysis. After determining the number of factors, we refit the EFA model using iterated principal factor analysis. We assumed that these factors might be correlated; therefore, oblique rotation (promax) was used. Items with uniqueness <0.5, highest loading on a factor <0.4, or high loadings (≥0.4) on >1 factor would be dropped. We also dropped factors that have <3 high-loading (loading ≥0.4) items. The process was iterated until no more items would be dropped from the model. The final model was examined for theoretical meaningfulness. We would report the result of Bartlett's test of sphericity and the Kaiser-Meyer-Olkin (KMO) measure of sampling adequacy to confirm the usefulness of EFA. We also report the internal consistency of the factors (using Cronbach's alpha). The attitude and practice scores were calculated using the factor loadings estimated in the final model. Similar to the knowledge score, these scores were also standardized with mean 0 and variance 1.

These knowledge, attitude, and practice scores were used as dependent variables in multiple linear regression models to determine the associated factors. The factors were chosen in accordance with a conceptual framework that was predicated on an assumption made about behavior. Before specific physician-targeted information on a health-related issue modifies patient outcome, it first affects physician's knowledge, then physician's attitude, and eventually physician's behaviour and practice [16]. (Fig 1).

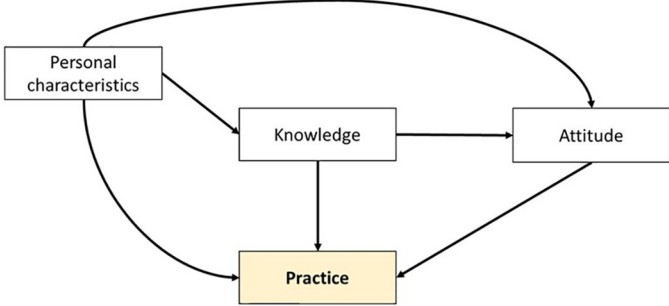

**Fig 1. Conceptual framework of factors related to the practice of TB care.**

## Ethical consideration

The study was conducted in accordance with the Declaration of Helsinki and Good Clinical Practice and was approved by the Institutional Review Board of Hanoi Medical University under decision No. 101/GCN-HĐĐĐNCYSSH-ĐHYHN dated October 25[th], 2021. Before proceeding with the survey, participants who completed the paper-based method provided written informed consent, whereas those who participated in the online method provided online informed consent. Each physician would receive 100,000 Vietnamese Dong as the compensation for their participation in this study. The investigators were responsible for protecting the privacy and confidentiality of participants as per Vietnam's regulations and Good Clinical Practice.

## Results

### Participant characteristics

A total of 232 physicians were included in this study, of which 53.9% were male and median age was 42 (IQR 36–47). Most were internal medicine (40.1%) or pulmonary physicians (30.6%). Around 80% reported being involved with TB in their work (76.3%), having examined TB patients (87.1%), having experience in examining TB patients (81%), and having prior TB training (78.9%) (Table 1).

### Knowledge

Most physicians had basic knowledge of TB, such as the cause of TB, common symptoms, screening and diagnostic tests, and the effectiveness of N95 mask (Table 2). However, important knowledge gaps were noted. Over 80% of physicians considered atypical manifestation (rash, nausea, vomiting, and headache) to be the main symptoms of TB. Only 20%–30% chose chest X-ray and IGRA to be the appropriate screening tests for TB while almost all chose urine test, complete blood count, and computed tomography of the chest. Only 17.2% knew that surgical mask was ineffective against TB.

### Attitude

Only 15%–20% of participants agreed with the statements regarding negative feelings towards TB patients. Nearly 70% expressed their willingness to participate in TB training. Only 20% agreed with the statements regarding the problems with the TB program. Around 70% believed in the benefits that the TB program could bring about (Fig 2).

The EFA suggested that three factors could adequately explain the variance observed in the attitude questionnaire. We named these factors "stigma towards TB" (6 items), "disbelief in the TB program" (6 items), and "motivation for TB care" (3 items). The factors had good internal consistency (Cronbach's alpha 0.93, 0.86, and 0.78, respectively). The Bartlett test of sphericity p-value was <0.0001 and the KMO measure of sampling adequacy was 0.91, suggesting that the EFA model was adequate.

In linear regression, mean "stigma" score was higher in physicians with higher degree and other physicians than primary care physicians, and lower in those with better knowledge. The "disbelief" attitude shared a similar picture, but knowledge appeared to play a more important role compared to other factors. Lower "motivation" attitude was also observed in these groups of physicians (S2 Table in S1 Appendix).

### Practice

Important practice gaps were identified in the TB practice. Chest X-ray and CT scan were usually requested to confirm TB diagnosis (~80%). HIV testing for confirmed active TB cases was

**Table 1. Participant characteristics (n = 232).**

| Characteristic | Results |
|---|---|
| Sex, n (%) | |
| Male | 107 (46.1) |
| Female | 125 (53.9) |
| Age (year), median (IQR) | 42.0 (36.0; 47.0) |
| Highest degree, n (%) | |
| Medical doctor | 158 (68.1) |
| Specialist level 1/Master | 64 (27.6) |
| Specialist level 2/PhD | 10 (4.3) |
| Specialty, n (%) | |
| Internal medicine | 93 (40.1) |
| Pulmonary medicine | 71 (30.6) |
| Primary care | 49 (21.1) |
| Family medicine | 19 (8.2) |
| Duration of work (year), median (IQR) | 15.0 (9.3; 20.0) |
| Directly involved with TB patients, n (%) | 177 (76.3) |
| Having seen TB patients at the clinic, n (%) | 202 (87.1) |
| Having experience in examining TB patients, n (%) | 188 (81.0) |
| Having prior TB training, n (%) | 183 (78.9) |
| Reasons for not having prior TB training (n = 49) | |
| Lack of time, n (%) | 27 (55.1) |
| Lack of information about the training | 41 (83.7) |
| Facility has separate rooms for TB, n (%) | 7 (15.9) |
| Reasons of not having separate rooms for TB (n = 170) | |
| Financial deficiency, n (%) | 71 (41.5) |
| Lack of space, n (%) | 96 (56.1) |
| Lack of equipment, n (%) | 96 (56.1) |
| Human resources deficiency, n (%) | 98 (57.3) |
| Inadequate infrastructure, n (%) | 125 (73.1) |
| Lack of TB diagnostics, n (%) | 113 (66.1) |

**Abbreviations**: IQR, interquartile range; TB, tuberculosis.

only requested by half of the participants. More than half of the participants reported that they often/usually/always engaged in these following inappropriate practices: sputum testing only for MDR-TB cases (65,1%), requesting follow-up examination only at the end of the treatment course (64.2%), and not ordering sputum smear at follow up examinations (53.8%). Instead of N95 mask, 86.2% of the participants reported that they often/usually/always wore a surgical mask when examining TB patients (Fig 3).

The EFA suggested that two factors could adequately explain the variance observed in the practice questionnaire. We named these factors "TB management" (4 items) and "TB precautions" (3 items). The factors had good internal consistency (Cronbach's alpha 0.85 and 0.91, respectively). The Bartlett test of sphericity p-value was <0.0001 and the KMO measure of sampling adequacy was 0.80, suggesting that the EFA model was adequate.

In linear regression, mean "TB management" score was lower in female physicians (mean difference -0.196, 95%CI -0.397, 0.004) and those with more "stigma" attitude (mean difference -0.311, 95%CI -0.473, -0.149). It was also lower in physicians with higher degree and other physicians than primary care physicians, but the differences were not statistically

**Table 2. Knowledge regarding TB (n = 232).**

| No. | Questions | Answer | Agreed with answer, n (%) |
|---|---|---|---|
| 1 | What is the cause of TB? | *Bacteria | 227 (97.8) |
| 2 | What is the most common transmission route? | *Airborne | 177 (76.3) |
| 3 | Who has the highest risk of TB? | *HIV patient | 142 (61.2) |
| 4 | Is latent TB contagious? | *No | 153 (65.9) |
| 5 | Is active TB contagious? | *Yes | 228 (98.3) |
| 6 | What are the main symptoms of TB? | *Cough ≥ 2 weeks | 227 (97.8) |
|  |  | *Mild fever | 211 (90.9) |
|  |  | Dizziness | 149 (64.2) |
|  |  | *Fatigue, weight loss | 162 (69.8) |
|  |  | Headache | 185 (79.7) |
|  |  | *Dyspnea, chest pain | 148 (63.8) |
|  |  | Nausea, vomiting | 197 (84.9) |
|  |  | *Night sweat | 157 (67.7) |
|  |  | Blurred vision | 33 (14.2) |
|  |  | Rash | 211 (90.9) |
| 7 | What is the diagnostic test for TB? | *Sputum culture | 170 (73.3) |
| 8 | What are the screening tests for TB? | *Mantoux | 198 (85.3) |
|  |  | Sputum culture | 162 (69.8) |
|  |  | Urine test | 229 (98.7) |
|  |  | *IGRA | 77 (33.2) |
|  |  | CBC | 226 (97.4) |
|  |  | *CXR | 53 (22.8) |
|  |  | CT chest | 214 (92.2) |
| 9 | How many samples are needed for sputum testing? | *2 samples | 184 (79.3) |
| 10 | Is surgical mask effective against TB? | *No | 40 (17.2) |
| 11 | Is N95 mask effective against TB? | *Yes | 227 (97.8) |
| 12 | What is the environmental control measure for TB? | *Natural ventilation | 169 (72.8) |

**Abbreviations**: TB, tuberculosis; HIV, human immunodeficiency virus; CXR, chest X-ray; IGRA, interferon-gamma release assays; CBC, complete blood count; CT, computed tomography; N95, non-oil 95.

*Correct answer based on the guidelines for TB diagnosis and treatment published by the Vietnam's Ministry of Health [15].

In linear regression, the mean knowledge score was higher in physicians with prior TB training (mean difference 1.009, 95%CI 0.764, 1.254) and lower in those with higher degree and shorter duration in current position and other physicians than primary care physicians (S1 Table in S1 Appendix).

significant. Mean "TB precaution" score was higher in female physician, but the difference was not statistically significant. It was lower in those with more "stigma" attitude (mean difference -0.242, 95%CI -0.433, -0.050). Better knowledge and "motivation" attitude were associated with better practice in terms of both TB management and precautions, while disbelief in TB program did not appear to be associated with any of the practice factors (Table 3).

## Discussion

In this study, we investigated the practice of TB care among physicians at private facilities. Our data provided several concerning results. We found that the participants were over-reliant in the imaging modalities, such as CXR or chest CT scan, to diagnose TB. According to the International Standards for Tuberculosis Care (ISTC) and the Vietnam Ministry of Health's

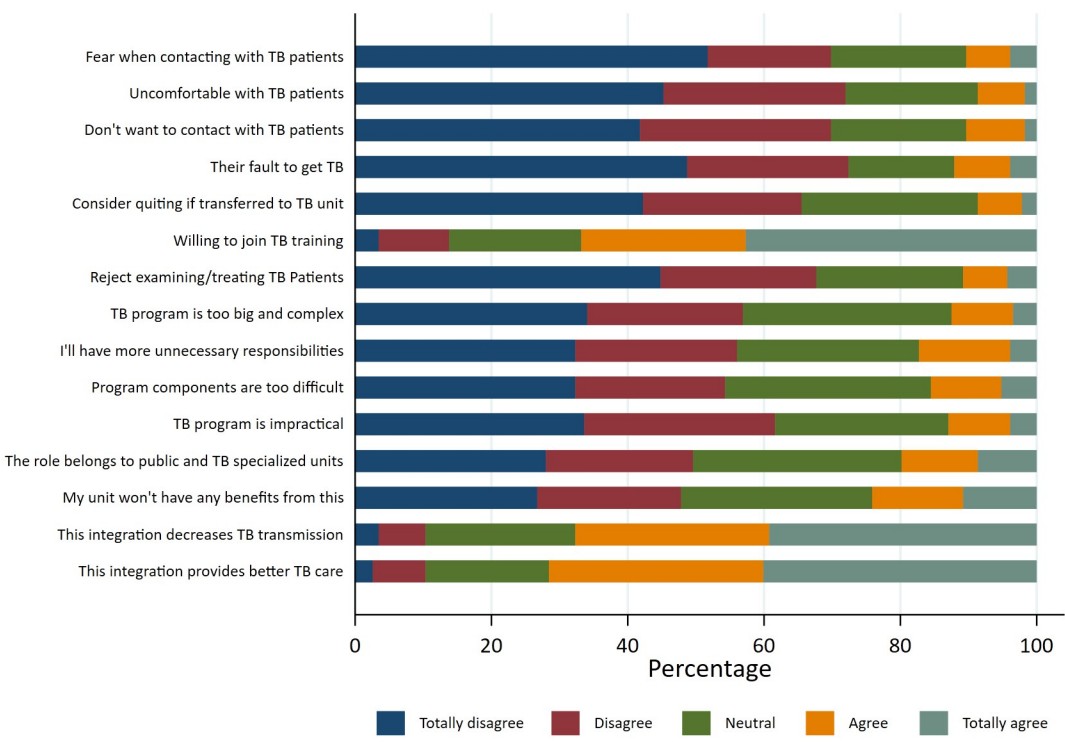

**Fig 2. Attitude towards TB patients and TB care program (n = 232).**

guidelines for TB diagnosis and treatment, CXR findings alone are unreliable for diagnosing TB, and sputum examination must be performed when CXR findings are suggestive [15, 17]. In other studies, CXR alone was the preferred investigations for the diagnosis of TB in 45,4%–68% of the participants [18–21]. This tendency could either be due to the lack of knowledge among private practitioners regarding the availability/reliability of sputum smear in TB diagnosis, or to the fact that the CXR/CT scan is faster and more convenient, making it more suitable for the practice in private sector, where private providers are frequently under pressure to provide immediate results to their clients.

Because HIV is the most important risk factor for developing active TB in the high-burden settings, the WHO has recommended HIV voluntary counseling and testing as a standard practice in patients with TB [22]. However, according to our findings, HIV testing for confirmed active TB cases was usually not done routinely. In a cross-sectional standardized-patient (SP) study conducted in 398 private providers in Nigeria, among presumptive and TB confirmed cases, HIV status was queried by only 13.4% and 27.1% of the providers, and HIV testing was recommended only by 10.3% and 13.4% of the providers [23]. In another SP study among 212 private practitioners in two South African cities, HIV status was queried in 24% of typical TB visits and 41% of confirmed TB visits [24]. Better practice was reported in a study on 540 HCWs from the public sector, as 74% of the participants ordered HIV tests for patients with confirmed TB. These findings, in general, emphasize the need for educating private physicians to increase their awareness and practice of ordering HIV testing for TB-confirmed patients.

Poor practice in treatment monitoring was noted, with the majority of participants reported that follow up examination was only often requested at the end of the treatment course, and sputum microscopy was not requested at follow up examinations. This gap was also mentioned

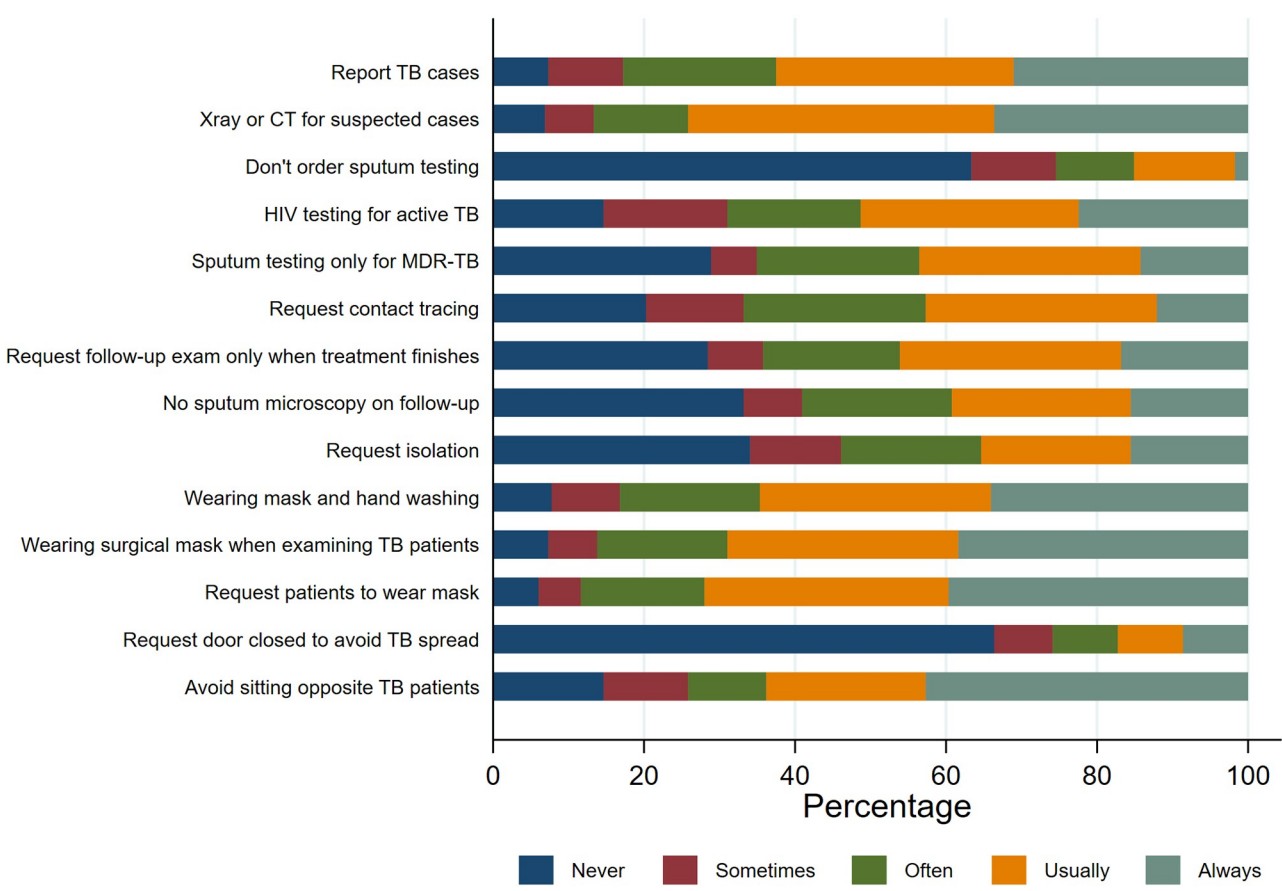

**Fig 3. Practice of TB management and precautions (n = 232).**

**Table 3. Factors associated with TB practice (n = 232).**

| Factors | TB management | TB precautions |
|---|---|---|
| Gender female | -0.197 (-0.398, 0.004) | 0.127 (-0.111, 0.365) |
| Duration in the current position | 0.013 (-0.000, 0.027) | **-0.024 (-0.039, -0.008)** |
| Highest degree (reference group: Medical doctor) | | |
| Specialist level 1/Master | -0.184 (-0.422, 0.054) | 0.269 (-0.013, 0.551) |
| Specialist level 2/PhD | -0.237 (-0.753, 0.278) | 0.009 (-0.602, 0.620) |
| Specialty (reference group: Primary care) | | |
| Internal medicine | -0.104 (-0.370, 0.162) | 0.095 (-0.220, 0.410) |
| Pulmonary medicine | -0.080 (-0.359, 0.199) | 0.181 (-0.149, 0.511) |
| Family medicine | 0.056 (-0.362, 0.473) | 0.143 (-0.351, 0.637) |
| Knowledge | **0.165 (0.035, 0.295)** | **0.165 (0.012, 0.319)** |
| Attitude | | |
| Stigma towards TB | **-0.312 (-0.474, -0.150)** | **-0.242 (-0.434, -0.050)** |
| Disbelief in the TB program | -0.093 (-0.248, 0.063) | 0.019 (-0.165, 0.204) |
| Motivation for TB care | 0.106 (-0.022, 0.234) | **0.195 (0.043, 0.346)** |

**Abbreviations**: TB, tuberculosis.

Coefficients in bold are statistically significant.

in previous studies. In a 2010 systematic review of knowledge, attitude, and practice among private providers, less than 20% of the participants provided follow up examinations for TB patients [8]. In a 2015 systematic review which included 47 studies, less than 40% of the providers were aware that sputum smear microscopy is required for monitoring response to treatment, while the remaining providers used clinical improvement and/or chest radiography to evaluate treatment response [25]. This is a deviation from standard practices, which requires supervision, patient support, as well as facility-based directly observed treatment (DOT) as per national treatment guidelines [15]. Appropriate follow-up and management of the TB patients ensures patient's compliance to treatment, since patients do have a tendency to disrupt the treatment because symptom improvement creates a pseudo impression that they have been cured. According to the ISTC and the national treatment guideline, sputum microscopy must be repeated after the intensive phase of treatment and after treatment completion, as treatment decision relies on sputum smear conversion result [15, 17]. Over-reliance on radiography could lead to inappropriate and unnecessary changes in treatment, as an unimproved X-ray finding within 3 months of treatment is not uncommon and is not a matter of concern. Ineffective practice in treatment monitoring and patients management would eventually lead to poor treatment adherence, resulting in treatment failure, disease relapse and drug resistance development [26].

Among the control measures recommended by the WHO to prevent TB transmission, personal respiratory protection by respirators is considered to be the most effective measure [27]. Several types of respirators have been widely used in healthcare settings, but only N95 respirator is recommended, by both the WHO and national treatment guideline, to prevent TB infection since it is able to protect the wearer from "inhaling airborne contaminants [15, 27–30]. However, 86.2% of the participants in our study preferred surgical mask to N95 respirator when examining TB patients. Recent studies have shown better practice in public sector, especially in MDR-TB settings, with around 40%–65% of the participants reporting/being observed to be using N95 respirators, while data in private sector was lacking [31–33]. This practice gap in our study could be explained by the lack of knowledge, since only 17.2% of the participants in our study knew that surgical mask was ineffective against TB. On the other hand, unavailability of respiratory protective equipment could limit their use in the hospital, as widespread and constant use of appropriate N95 respirators is regarded as impractical in private setting, due to the financial constrains.

In our study, having prior TB training was associated with better knowledge and less stigmatizing attitude, while both of these factors were associated with better practice in terms of TB management and precautions. These findings suggest that adequate knowledge and positive attitude could translate into appropriate practices of HCWs on TB care, and by enhancing their knowledge and attitude, TB training would consequently improve their practice. In a 2022 qualitative study of barriers to TB case finding in primary and secondary health facilities in Ghana, lack of TB knowledge and lack of training on case detection guidelines were listed as major HCW-related barriers, as they could result in missed or delayed diagnosis of TB [34]. In a 2014 qualitative study which included 76 HCWs, lack of knowledge and stigmatizing attitude towards TB patients among HCWs on TB management were thought to have negative effects on patient's compliance to TB treatment [35]. Previous studies have demonstrated a correlation between knowledge and attitude with the practice of HCWs in TB precautions [36–39], however, as also indicated by other studies, there was no association between knowledge, attitude, and practice of HCWs regarding TB management [13, 40, 41]. Our results highlighted the need of specifically designed and tailored training programs which not only imparts adequate knowledge but also helps build up more appropriate attitudes and improve the practice of TB among HCWs in the private sector.

To our knowledge, this is the first study evaluating the knowledge, attitude, and practice regarding TB care among Vietnamese private providers. The questionnaires used in this study have good psychometric properties and have been shown to explore latent constructs regarding attitude and practice. However, we could not measure actual practice by observing or by using the simulated client method. Since self-report practice tends to be better than actual practice (the know-do gap) [23, 24], there may have been an overestimation of appropriate practice among the HCWs. Furthermore, part of our study participants were physicians working in health facilities in the Friends for International TB Relief (FIT) network, who may have previously received training and ongoing support from FIT's programs, potentially leading to improved practices in TB care and introducing bias into our results. However, these biases will not change the current situation described in this paper.

## Conclusions

There were important gaps in knowledge, attitude, and practice of TB care among private providers. Better knowledge was associated with positive attitude towards TB and better practice. Tailored training may help to address these gaps and improve the quality of TB care in the private sector.

## Supporting information

**S1 Appendix.**
(DOCX)

## Author Contributions

**Conceptualization:** Do Minh Ngo, Son Nam Tran, Long Bao Hoang, Hoa Binh Nguyen.

**Data curation:** Do Minh Ngo, Ngoc Bao Doan, Long Bao Hoang.

**Formal analysis:** Do Minh Ngo, Ngoc Bao Doan, Long Bao Hoang.

**Investigation:** Do Minh Ngo, Ngoc Bao Doan.

**Methodology:** Do Minh Ngo, Ngoc Bao Doan, Son Nam Tran, Long Bao Hoang, Hoa Binh Nguyen, Vung Dang Nguyen.

**Project administration:** Hoa Binh Nguyen, Vung Dang Nguyen.

**Supervision:** Do Minh Ngo, Long Bao Hoang, Hoa Binh Nguyen, Vung Dang Nguyen.

**Validation:** Do Minh Ngo, Hoa Binh Nguyen, Vung Dang Nguyen.

**Visualization:** Ngoc Bao Doan, Long Bao Hoang, Hoa Binh Nguyen, Vung Dang Nguyen.

**Writing – original draft:** Do Minh Ngo, Ngoc Bao Doan, Son Nam Tran.

**Writing – review & editing:** Do Minh Ngo, Ngoc Bao Doan, Long Bao Hoang, Hoa Binh Nguyen, Vung Dang Nguyen.

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
