## [Decision Letter · Decision Letter 0]

20 Mar 2023

PONE-D-23-02304ORIGINAL RESEARCH

Practice regarding tuberculosis care among physicians at private facilities: A cross-sectional study from VietnamPLOS ONE

Dear Dr. Ngo

Thank you for submitting your manuscript to PLOS ONE. After careful consideration, we feel that it has merit but does not fully meet PLOS ONE’s publication criteria as it currently stands. Therefore, we invite you to submit a revised version of the manuscript that addresses the points raised during the review process. Two reviewers have made comments which I urge you to take into account when revising your manuscript.

We look forward to receiving your revised manuscript.

Kind regards,

Yogan Pillay, Phd

Academic Editor

PLOS ONE

Journal Requirements:

Additional Editor Comments:

Many thanks for your manuscript which as you point out is the first on this subject. Both reviewers found the paper interesting but require some revision as indicated below. Kindly ensure that the revised manuscript is edited for language.

Reviewer 1:

In the abstract, Line 24 Results Section please be clear if you are referring to HIV positive individuals or confirmed TB testing for HIV presence.

Line 65 : what do you mean legal expertise of physicians

Line 67 and 68: grammar needs correction

Line 71 not 'on'- 'at'

Eligibility

Can you clarify if in Vietnam there are TB specific physicians as you suggest in line 79. If so, they should be described somewhere in the paper. Their roles generally, distribution, skill level etc

Sampling

To exclude probabilistic sampling, the author should at least describe how many clinicians there were in the regions focused on and the time constraint.

who is FIT and what was their role and who belongs to their network? Kindly exclude bias or note it if it exists.

Just to be clear, you only included those who completed the survey? So there was no follow up with those who didn’t?

Line 91 Survey Design

Is this validated or not?

Data Collection

Can you clarify that the paper and online methods were standardized and completion rates and other crtieria were similar.

Line 118- Grammar correction needed

Line 126- Reference use of IRT to estimate knowledge score

Results

Can you define at which level of care the physicians were? Primary or hospital based

Please describe the difference of 'directly involved with TB patients' and Having seen TB patients

The knowledge gaps are really large and should be i)disaggregated at least internally among different providers and ii) possibly to other studies of knowledge of TB amongst physicians.

which urine or blood test were being alluded to?

Discussion

It would be valuable to understand whether the 'threshold' for prompt to use imaging was appropriate ie. Symptoms that suggested TB or another screening modality.

Conclusion

Is there any policy implication of this work and how did the Department of Health see these findings

Reviewer 2:

Overall, the study has a good design, data analysis and interpretation. However, the conclusion about provision of training may be too simplistic. Available literature shows that often trained providers do not follow guidelines unless there are systems in place to ensure provider compliance. The issue in Viet Nam will be how to follow up on this in the private sector.

As noted by the authors, this is one of the first studies in Vietnam to highlighted KAP among private healthcare providers. The findings are critical for designing programmatic strategies to better engage private providers in TB care. The authors should provide data, if available, about percent of TB cases seen/notified by private providers. Also, percentage of sick people who visit private providers for seeking care as their first contact should be presented in the background section of the paper , if such data available. Viet Nam in 2021, notified approximately 46 percent of its TB cases. The authors note on page 3 that 50 to 70% of TB cases choose private pharmacies and clinics as their initial treatment point. Does this mean, these are the facilities that notify these cases to NTP? Some clarity is needed on this point.

Reviewers' comments:

Reviewer's Responses to Questions

**Comments to the Author**

1. Is the manuscript technically sound, and do the data support the conclusions?

Reviewer #1: Yes

Reviewer #2: Yes

2. Has the statistical analysis been performed appropriately and rigorously? 

Reviewer #1: Yes

Reviewer #2: I Don't Know

3. Have the authors made all data underlying the findings in their manuscript fully available?

Reviewer #1: Yes

Reviewer #2: Yes

4. Is the manuscript presented in an intelligible fashion and written in standard English?

Reviewer #1: Yes

Reviewer #2: No

5. Review Comments to the Author

Reviewer #1: Overall, the study has a good design, data analysis and interpretation. However, the conclusion about provision of training may be too simplistic. Available literature shows that often trained providers do not follow guidelines unless there are systems in place to ensure provider compliance. The issue in Viet Nam will be how to follow up on this in the private sector.

As noted by the authors, this is one of the first studies in Vietnam to highlighted KAP among private healthcare providers. The findings are critical for designing programmatic strategies to better engage private providers in TB care. The authors should provide data, if available, about percent of TB cases seen/notified by private providers. Also, percentage of sick people who visit private providers for seeking care as their first contact should be presented in the background section of the paper , if such data available. Viet Nam in 2021, notified approximately 46 percent of its TB cases. The authors note on page 3 that 50 to 70% of TB cases choose private pharmacies and clinics as their initial treatment point. Does this mean, these are the facilities that notify these cases to NTP? Some clarity is needed on this point.

Reviewer #2: In the abstract, Line 24 Results Section please be clear if you are referring to HIV positive individuals or confirmed TB testing for HIV presence.

Line 65 : what do you mean legal expertise of physicians

Line 67 and 68: grammar needs correction

Line 71 not 'on'- 'at'

Eligibility

Can you clarify if in Vietnam there are TB specific physicians as you suggest in line 79. If so, they should be described somewhere in the paper. Their roles generally, distribution, skill level etc

Sampling

To exclude probabilistic sampling, the author should at least describe how many clinicians there were in the regions focused on and the time constraint.

who is FIT and what was their role and who belongs to their network? Kindly exclude bias or note it if it exists.

Just to be clear, you only included those who completed the survey? So there was no follow up with those who didnt?

Line 91 Survey Design

Is this validated or not?

Data Collection

Can you clarify that the paper and online methods were standardized and completion rates and other crtieria were similiar.

Line 118- Grammer correction needed

Line 126- Reference use of IRT to estimate knowledge score

Results

Can you define at which level of care the physicians were? Primary or hospital based

Please describe the difference of 'directly involved with TB patients' and Having seen TB patients

The knowledge gaps are really large and should be i)disaggregated at least internally among different providers and ii) possibly to other studies of knowledge of TB amongst physicians.

which urine or blood test were being alluded to?

Discussion

It would be valuable to understand whether the 'threshold' for prompt to use imaging was appropriate ie. Symptoms that suggested TB or another screening modality.

Conclusion

Is there any policy implication of this work and how did the Department of Health see these findings

6. PLOS authors have the option to publish the peer review history of their article (what does this mean?). If published, this will include your full peer review and any attached files.

Reviewer #1: No

Reviewer #2: No

---

## [Author Response · Author response to Decision Letter 0]

23 Mar 2023

Do Minh Ngo, MSc.

National TB control Program

National Lung Hospital

No. 463, Hoang Hoa Tham, Ba Dinh, Hanoi, Vietnam

Email: minhdotb@gmail.com

Yogan Pillay, PhD

Academic Editor

PLOS ONE

March 22nd, 2023

Subject: Revision and resubmission of the manuscript ID: PONE-D-23-02304, title: "Practice regarding tuberculosis care among physicians at private facilities: A cross-sectional study from Vietnam" 

Dear Dr Yogan Pillay,

Thank you for giving me the opportunity to submit a revised draft of my manuscript titled "Practice regarding tuberculosis care among physicians at private facilities: A cross-sectional study from Vietnam" to the Plos One Journal. The suggestions offered by the reviewers have been immensely helpful, and we also appreciate your insightful comments on revising the abstract and other aspects of the paper.

I have included the reviewer comments immediately after this letter and responded to them individually, indicating exactly how we addressed each concern or problem and describing the changes we have made. The revisions have been approved by all authors and I have again been chosen as the corresponding author. The changes have been highlighted within the manuscript, and the revised manuscript is attached to this email message. A point-by-point response to the reviewers’ comments and concerns is presented in the next page.

We would like to express our sincere and heartfelt gratitude to the reviewer for their insightful and constructive comments on our manuscript. Your feedback has been invaluable in improving the quality and clarity of our work, and we appreciate the time and effort you have taken to carefully review our submission. Your feedback has not only strengthened the scientific merit of our study but also inspired us to pursue further research in this field. Once again, thank you for your invaluable contribution to our work. We look forward to hearing from you in due time regarding our submission and to respond to any further questions and comments you may have.

Sincerely,

Do Minh Ngo, MSc.

National TB control Program

National Lung Hospital

No. 463, Hoang Hoa Tham, Ba Dinh, Hanoi, Vietnam

Email: minhdotb@gmail.com

 

RESPONSES

Reviewer 1

Comment 1: In the abstract, Line 24 Results Section please be clear if you are referring to HIV positive individuals or confirmed TB testing for HIV presence.

Response: The aforementioned text referred to the protocol of ordering HIV testing in TB confirmed cases. Given that HIV is the most important risk factor for TB, especially in high burden setting, WHO and NTP guidelines have recommended HIV voluntary counseling and testing as a standard practice for patients with confirmed TB diagnosis. Additional information concerning this topic is detailed in the Discussion section, paragraph 2, lines 252-263.

Comment 2: Line 65: What do you mean legal expertise of physicians?

Response: According to Article 5 of Circular No. 36/2021/TT-BYT on the regulations regarding the scope of specialization in TB care, which only be provided by (a) clinical physicians who specialize in TB or pulmonology; or (b) clinical physicians who are not, but have received training in TB care in accordance with the national guidelines and have been certified by the NTP. We apologize for any confusion caused and have rectified this oversight in the revised manuscript. The relevant circular has also been cited in the revised lines, which can be found at lines 65-67 and have been highlighted for your convenience.

Comment 3: Line 67 and 68: grammar needs correction.

Response: We have corrected the errors.

Comment 4: Line 71 not 'on'- 'at'

Response: We have corrected the errors.

Comment 5: Can you clarify if in Vietnam there are TB specific physicians as you suggest in line 79. If so, they should be described somewhere in the paper. Their roles generally, distribution, skill level etc

Response: In Vietnam, the number of physicians specializing in TB is limited, with majority of them working in TB-specific hospitals or departments in NTP network or public facilities. Private sector TB physicians are rare and typically retired physicians from the public sector who have relocated to private facilities. As they have already acquired the necessary knowledge and skills for TB care in accordance with national guidelines, they are not the primary target for this study or for future interventions. We highly appreciate the suggestion of mentioning TB physicians in the paper, so the readers can have a better overall picture of TB care in Vietnam; however, we do not have an exact figure of their number, skill level, and other information. Future studies that examine TB care in Vietnam comprehensively will be more appropriate to integrate this group of physicians.

Comment 6: To exclude probabilistic sampling, the author should at least describe how many clinicians there were in the regions focused on and the time constraint.

Response: Current data regarding the exact number of private TB doctors in the entire nation is not available. Unlike the public sector where hospitals are often connected through a network of training, education, and professional direction, the private sector is harder to reach; therefore, contacting every private clinic is impossible. This is what we meant by “time and resources constraint”. Even if we were able to contact every clinic, response rate would be very low because they were not familiar with the research team. That is why we utilized a network of private TB clinics instead.

Comment 7: Who is FIT and what was their role and who belongs to their network? Kindly exclude bias or note it if it exists.

Response: The Friends for International TB Relief (FIT) is a non-profit organization that focuses on improving the health of communities affected by tuberculosis (TB) around the world. FIT operates in several countries, including Vietnam, and works with local partners to implement TB prevention, diagnosis, and treatment programs. In Vietnam, FIT works closely with the National TB Program and other local organizations to support the fight against TB. Their efforts in Vietnam include providing technical assistance, training for healthcare workers, and conducting research on TB prevention and treatment. Additionally, FIT has contributed to the development of new TB diagnostic and treatment tools and has helped to build capacity for TB control in the country.

We appreciate your input regarding the potential bias in our study concerning physicians affiliated with health facilities in the Friends for International TB Relief (FIT) network. It is important to note that these physicians have previously received training from FIT's programs, in addition to ongoing support such as technical assistance, diagnostic and treatment tools, and case notification assistance. Consequently, it is possible that they may have better practices in TB care, which could introduce bias into our findings. We have included this limitation in our paper as noted in lines 321-325 of the final paragraph of the Discussion section.

Comment 8: Just to be clear, you only included those who completed the survey? So there was no follow up with those who didn’t?

Response: Yes, in this study we only included physicians who completed the survey. The research team wanted to anonymize the responders by not asking them for their names and phone numbers in the online survey form (see lines 111-112); therefore, it’s impossible to reidentify those with missing data.

Comment 9: Line 91 Survey Design. Is this validated or not?

Response: The questionnaires have not been validated elsewhere. Therefore, in this study, we performed latent variable analyses to examine the structure of the questionnaires; for example, we used exploratory factor analysis to examine the structure of the Attitude and Practice questionnaires (see lines 134-151). As described in the manuscript, we did not use the entire questionnaire but only selected items that appeared to be valid for the constructs. We also examined some psychometric properties of the constructs. For example, we examined the internal consistency of these constructs using Cronbach's alpha. We have described the validation analyses in the Results section, specifically in lines 200-205 and lines 221-225.

Comment 10: Can you clarify that the paper and online methods were standardized and completion rates and other crtieria were similar.

Response: Questions were developed by a team including epidemiologists and TB doctors who have experiences in deploying questionnaires. We then piloted the questionnaire, both paper and online, with some colleagues and refined the language. Only a few online responders did not complete the online survey.

Comment 11: Line 118- Grammar correction needed

Response: We have corrected the errors.

Comment 12: Line 126- Reference use of IRT to estimate knowledge score

Response: The item response theory aims to estimate a latent trait θ, which is assumed to be continuous. The knowledge score is indeed the estimated (predicted) location of a person on the scale of the latent trait θ (“knowledge”). This scale has been standardized to a standard normal distribution (mean 0 and variance 1); for example, people with knowledge scores ≥2 are the 2.5% of the population with the best knowledge. Although not reported, the number of correct answers was highly correlated with the knowledge score; however, we felt that standardizing the scale provides an easier measure to compare, especially when we include the variable in other models.

Comment 13: Can you define at which level of care the physicians were? Primary or hospital based?

Response: The study was carried out among physicians operating in clinics, thereby characterizing the level of service as primary care.

Comment 14: Please describe the difference of 'directly involved with TB patients' and Having seen TB patients.

Response: "Directly involved with TB patients" typically refers to physicians who have frequent and close contact with patients who are diagnosed with tuberculosis (TB), such as pulmonary physicians. They may be involved in various aspects of TB patient care, including diagnosis, treatment, and monitoring. "Having seen TB patients" generally implies that the person has come across TB patients in their professional or personal life, but may not necessarily have direct involvement in their care. The key difference between these two terms is the level and nature of involvement with TB patients. "Directly involved" suggests more hands-on involvement with patient care, whereas "having seen" implies a more passive or incidental encounter with TB patients.

Comment 15: The knowledge gaps are really large and should be i) disaggregated at least internally among different providers and ii) possibly to other studies of knowledge of TB amongst physicians.

Response: We concur with this observation as there were substantial gaps in knowledge, and we'd like to point out that the knowledge regarding TB care have been disaggregated internally among different providers in this study, specifically in the multilinear regression model of knowledge score in Table S3 of the Appendix. Additionally, since the primary aim of this study was the practice regarding TB care, we focus on comparing the practice gap, rather than the knowledge gap of our study with other study in the Discussion section.

Comment 16: Which urine or blood test were being alluded to?

Response: In this study, urine test was referred to urinalysis, whereas blood tests were comprised of complete blood count and comprehensive metabolic panel.

Comment 17: It would be valuable to understand whether the 'threshold' for prompt to use imaging was appropriate ie. Symptoms that suggested TB or another screening modality.

Response: Based on Vietnam’s national guidelines, chest X-rays are recommended for individuals suspected of having tuberculosis, as suggested by persistent cough, mild fever, dyspnea, chest pain, and night sweats, or for those at high risk, such as those who are HIV-positive. However, it is important to note that in the study, inappropriate practices regarding chest X-ray orders occurred when TB diagnosis was solely based on the results of the chest X-ray or other image modalities, without considering other microbiological evidence such as sputum, genXpert, or culture. This inappropriate practice relates to how physicians use chest X-rays in making a diagnosis, rather than to the specific individuals for whom chest X-rays are ordered.

Comment 18: Is there any policy implication of this work and how did the Department of Health see these findings

Response: This is an important question and thanks for asking that. The primary investigator of this study works at the National Lung Hospital, the highest level of professional direction. Therefore, our next step is to present this to the Director Board as well as the Ministry of Health, through meetings and conferences, so that they are aware of the situation. Our findings, despite being preliminary and not making any causal implications, will fill in an important gap in the understanding of how the private sector can contribute to TB care in Vietnam. These finding can also be included in future grant applications to secure more funding for training and activities to improve the awareness and attitude for private TB practitioners.

Reviewer 2

Comment 1: Overall, the study has a good design, data analysis and interpretation. However, the conclusion about provision of training may be too simplistic. Available literature shows that often trained providers do not follow guidelines unless there are systems in place to ensure provider compliance. The issue in Viet Nam will be how to follow up on this in the private sector.

Response: We concur entirely with the viewpoint of the reviewer. In addition to training provision, other control measures must be enforced by policy. However, those measures were not studied in this study; therefore, we did not make any conclusion beyond our findings. Moreover, the situation reported in this study suggests that private TB practitioners do not have adequate knowledge. If we were to enforce other measures, we still need start with providing them with standard knowledge.

Comment 2: As noted by the authors, this is one of the first studies in Vietnam to highlighted KAP among private healthcare providers. The findings are critical for designing programmatic strategies to better engage private providers in TB care. The authors should provide data, if available, about percent of TB cases seen/notified by private providers. Also, percentage of sick people who visit private providers for seeking care as their first contact should be presented in the background section of the paper , if such data available. Viet Nam in 2021, notified approximately 46 percent of its TB cases. The authors note on page 3 that 50 to 70% of TB cases choose private pharmacies and clinics as their initial treatment point. Does this mean, these are the facilities that notify these cases to NTP? Some clarity is needed on this point.

Response: We concur with your statement about the contributions of private healthcare systems to the notification cases of TB in Vietnam. Nonetheless, there is currently no official data available on this subject, and obtaining TB-related data from private healthcare facilities in Vietnam is challenging. The majority of TB data is self-reported and falls short of standards, primarily due to private healthcare providers not recording TB information, as they offer patients "unofficial" TB-related services. Additionally, several factors, such as stigmatization, confidentiality of sensitive information, including financial details, and legal considerations regarding the standards of private healthcare providers and professional certificates for healthcare workers in TB care, hinder accessing TB data and private healthcare facilities.

The statistical evidence indicating that a significant proportion, ranging between 50% and 70%, of TB patients seek treatment from private pharmacies and clinics as their initial care point was derived from three cross-sectional studies conducted in Vietnam (References 9-11). Notably, these studies did not rely on notifications emanating from the private sector. Rather, the authors conducted surveys among confirmed TB cases that included inquiries regarding the location where patients sought medical assistance upon onset of initial symptoms.

Additional requirements

Comment 1: Please ensure that your manuscript meets PLOS ONE's style requirements, including those for file naming.

Response: We have reviewed the guidelines and made the necessary adjustments to comply with the recommended formatting. The revised manuscript now adheres to PLOS ONE's style templates, and all files have been appropriately named.

Comment 2: Please provide additional details regarding participant consent. In the ethics statement in the Methods and online submission information, please ensure that you have specified what type you obtained (for instance, written or verbal, and if verbal, how it was documented and witnessed). If your study included minors, state whether you obtained consent from parents or guardians. If the need for consent was waived by the ethics committee, please include this information.

Response: Thank you for raising the concern regarding participant consent. In our revised manuscript, we have included additional information on the type of consent obtained in the Ethical Consideration section, lines 163-166. For your convenience, these changes have been highlighted.

Comment 3: Please include captions for your Supporting Information files at the end of your manuscript, and update any in-text citations to match accordingly.

Response: We appreciate the reviewer's suggestion to include captions for our Supporting Information files at the end of our manuscript. In response, we have included captions for all Supporting Information files and updated any in-text citations to match accordingly. We have also reviewed PLOS ONE's Supporting Information guidelines and made sure that our files comply with the recommended format.

Comment 4: Please review your reference list to ensure that it is complete and correct. If you have cited papers that have been retracted, please include the rationale for doing so in the manuscript text, or remove these references and replace them with relevant current references. Any changes to the reference list should be mentioned in the rebuttal letter that accompanies your revised manuscript. If you need to cite a retracted article, indicate the article’s retracted status in the References list and also include a citation and full reference for the retraction notice.

Response: Thank you for bringing our attention to the need to review our reference list for completeness and accuracy. We have carefully reviewed the references and ensured that they are up-to-date and accurately cited in the manuscript.

---

## [Editor Report · Decision Letter 1]

4 Apr 2023

Practice regarding tuberculosis care among physicians at private facilities: A cross-sectional study from Vietnam

PONE-D-23-02304R1

Dear Dr. Ngo

We’re pleased to inform you that your manuscript has been judged scientifically suitable for publication and will be formally accepted for publication once it meets all outstanding technical requirements.

Kind regards,

Yogan Pillay, Phd

Academic Editor

PLOS ONE
---

## [Editor Report · Acceptance letter]

19 Apr 2023

PONE-D-23-02304R1 

Practice regarding tuberculosis care among physicians at private facilities: A cross-sectional study from Vietnam 

Dear Dr. Ngo:

I'm pleased to inform you that your manuscript has been deemed suitable for publication in PLOS ONE. Congratulations! Your manuscript is now with our production department. 

Kind regards, 

on behalf of

Dr. Yogan Pillay 

Academic Editor

PLOS ONE